# The Effect of a Low-Cost Body Weight-Supported Treadmill Trainer on Walking Speed and Joint Motion

**DOI:** 10.3390/medicina55080420

**Published:** 2019-07-30

**Authors:** Jessica D. Ventura, Ann L. Charrette, Katherine J. Roberts

**Affiliations:** 1Kinesiology Department, Gordon College, Wenham, MA 01984, USA; 2School of Physical Therapy, MCPHS University, Worcester, MA 01608, USA; 3Teachers College, Columbia University, New York, NY 10027, USA

**Keywords:** rehabilitation, gait training, assistive technology, brain injury

## Abstract

*Background and Objectives:* Gait training with body weight-support has been shown to improve the walking speed of individuals with movement disorders. The AccesSportAmerica Gait Trainer is a low-cost, pre-market gait rehabilitation device that alters the stride characteristics of participants walking on a standard treadmill. The purpose of this study was to examine the biomechanical outcomes that training on this device has for people with brain injuries that affect motor functioning. It was hypothesized that there would be an increase in walking speed post-intervention, and that there would be an increase in step length and joint range-of-motion. *Materials and Methods:* An intervention study was conducted with 11 people with ambulatory difficulty caused by post-stroke hemiparesis (*n* = 7), traumatic brain injury (*n* = 3), and cerebral palsy (*n* = 1). The average time using the AccesSportAmerica Gait Trainer was 34.5 (SD = 6.0) minutes per session for 36.9 (SD = 21.8) sessions. Gait speed, step length and time, and joint flexion were measured during the 10 Meter Walk Test. *Results:* From pre- to post-intervention, there was a mean increase in walking speed of 0.19 m/s (SD = 0.06, *p* = 0.016, *d* = 0.40) and a decrease in step time of both affected and unaffected legs (affected: *p* = 0.011, *d* = 0.37; unaffected: *p* = 0.004, *d* = 0.67). There was no significant change in stride length or joint angles. *Conclusions*: The AccesSportAmerica Gait Trainer has the potential to improve the walking speed of people with ambulatory difficulty.

## 1. Introduction

Over 20 million Americans live with ambulatory difficulty, defined as difficulty walking or climbing stairs [1]. The ability to walk is key to independent living and community dwelling. As a valid, reliable, and sensitive measure, walking speed is the functional vital sign used extensively by therapists to track changes in overall health and quality of life [2]. Gait speed as measured by the Ten Meter Walk Test (10MWT) is a simple assessment tool that provides a wealth of information about the ability to function in the home and community. There is considerable evidence that the 10MWT is an effective clinical tool to assess and monitor new interventions among a variety of populations with ambulatory difficulty [3,4,5,6].

Body weight-support treadmill training is an approach to improving gait speed and other gait parameters in individuals living with movement disorders [7,8,9]. Treadmill training provides therapists the ability to repeatedly manipulate joint motion during precise phases of the gait cycle while the patients are supported by a harness system. Manual methods for body weight-support treadmill training require hand-over-hand manipulation by therapists who are sitting on the side of the treadmill. These methods are staff intensive and lead to fatigue and imprecise correction of gait [10]. New technologies, categorized as exoskeletons and end-effector devices, have been developed to replace manual methods with robotic devices [11]. One example of a commercially available product is the Lokomat, a robotic gait orthosis combined with a harness-supported body weight system used to obtain correct form during training on a treadmill [12]. The Lokomat can operate up to 1.9 mph when using the gait orthosis [13]. Another example is the HapticWalker, a walking simulator with two programmable foot platforms that have permanent foot–machine contact [14]. The device simulates different types of terrains, including level floors, stairs, and walking terrain, and is able to recreate speeds up to 3 mph at 120 steps/min. The greatest obstacle to the widespread use of robotic devices is the high capital costs [15], limiting patients’ access to these devices. For this reason, Lilley [16] invented a low-cost alternative to devices currently on the market: the AccesSportAmerica Gait Trainer. 

Installed posterior to a standard treadmill with an overhead harness system, the AccesSportAmerica Gait Trainer guides users through the walking cycle by effectively linking the external motion of the treadmill belt into the body’s own kinetic chain (Figure 1) [16]. Support and precise motion are delivered to the lower leg and foot through a boot that is attached to a flywheel via fully adjustable shafts. The micro adjustments available on the AccesSportAmerica Gait Trainer allow for precise fitting to the participant’s size and easy variation of stride length, hip and knee flexion, ankle rotation, and heel-strike angle. The angular momentum contributed by the coupled flywheel facilitates smooth continuous motion through the entire gait cycle. The device requires only one monitoring trainer who can also pedal the stationary bicycle to influence gait, if necessary. The AccesSportAmerica Gait Trainer speeds can vary from 1.2 to 2 mph or from 2 to 6 mph, depending on the stride length. The training device is versatile, as it allows users to walk and even run safely. Participants can train for over an hour at a time for benefits of aerobic conditioning and improving biomechanical function. The AccesSportAmerica Gait Trainer avoids costly electronics, hydraulics, and other complex systems by employing a simple, robust design. Due to its simplicity, the device may be produced at lower cost, making it affordable to more facilities and accessible to a greater number of persons living with ambulatory difficulty.

The purpose of this study was to determine the effects that training on the AccesSportAmerica Gait Trainer has on walking speed and joint range of motion of participants during overground walking. It was hypothesized that the AccesSportAmerica Gait Trainer would improve gait speed as measured by the 10MWT as well as joint range of motion in individuals with brain injury. 

## 2. Materials and Methods

### 2.1. Participants

Sixteen participants were recruited from a larger group of individuals enrolled in the Gait Trainer Program offered annually by AccesSportAmerica (Acton, MA, USA) [17]. Inclusion criteria included permission from a doctor to exercise and the ability to bear one’s own body weight. Three participants did not complete the program due to schedule conflicts, and two dropped out because the training was too rigorous. The 11 participants (10 male; 1 female) who completed the intervention ranged in age from 24 to 71 years, with an average age of 44.8 (SD = 16.5) years. They had a variety of disabilities that affect motor function: seven were post-stroke with hemiparesis, three with traumatic brain injury (TBI), and one with cerebral palsy (CP). All conditions presented with high tone and affected the motor control of the extremities and trunk. The conditions were chronic (persistent, long-lasting health conditions) as the study participants had been living with their injuries for an average of 12.1 (SD = 7.0) years. When pathologies were bilateral, the weaker of the two limbs was defined as the affected limb for purpose of analysis. This study was approved by Gordon College Institutional Review Board (issued on 11 September 2017, No. FA-170404). All participants were informed of the rationale for the study and provided informed consent.

### 2.2. Intervention

Those enrolled in this study used the AccesSportAmerica Gait Trainer for an average of 36.9 (SD = 21.8) sessions over a 12- to 40-week period (Table 1). The treadmill speed and amount of time spent on the AccesSportAmerica Gait Trainer varied at the discretion of the attending therapist. In addition to 20 min or more of walking assistance, each session typically involved from 2 to 5 intervals of 1 to 2 min at a faster speed and/or with the treadmill in reverse. 

### 2.3. Gait Analysis

Gait speed was captured using the 10MWT. For the 10MWT, any ambulation aids such as a cane, orthoses, or assistant were consistent throughout the tests (Table 2). The participant walked along a straight, level 10-meter walkway. Participants were instructed to walk as fast as they safely could along the assessment distance [18]. To calculate walking velocity, the distance covered was divided by the time it took the individual to walk the 6 m in the center of the walkway, allowing 2 m each for acceleration and deceleration. The test–retest reliability of the 10MWT has been found to be excellent (intraclass correlation coefficient = 0.96) for fast speeds in people with stroke [4]. Concurrent validity has been examined in people following stroke with excellent correlation with the Barthel Index (*r* = 0.78). A minimal detectable change (MDC) in performance of greater than 0.05 m/s was found to exceed rater error in a study of people with TBI [19]. An increase in score on the 10MWT is indicative of improved ability to ambulate safely at home or in the community [2]. 

Using a webcam on a trolley, video was obtained during three trials pre- and post-intervention, and those videos were analyzed using Dartfish Pro-suite software (Fribourg, Switzerland) in order to get information about spatiotemporal changes and walking mechanics. A single stride was chosen from each video based on video quality and visibility of the joints. Step length and time were determined from foot strike of the contralateral leg to foot strike of the ipsilateral leg, with distances being measured from the heel of the shoe. Hip, knee, and ankle angles were defined as zero when the participant was in quiet stance, with extension or plantarflexion of the joint angles from the starting position defined as positive and flexion or dorsiflexion defined as negative. Ankle dorsiflexion (DF) was measured at contralateral limb foot strike; ankle plantarflexion (PF), knee flexion, and hip extension were measured at toe-off; and hip flexion was measured at foot strike. These events were chosen because they are close to the timing of peak joint angles during the stance phase of healthy walking [20].

### 2.4. Data Analysis

The means, standard deviations, and frequencies were calculated to describe the number of weeks in the program, total number of sessions, and average time on the AccesSportAmerica Gait Trainer per session. Walking velocity and joint angles were averaged across trials for pre- and post-intervention. Statistical analyses were performed using the software package SPSS 25.0. Normality of the data was tested using the Shapiro–Wilk test. A paired Student’s *t*-test was used to compare differences pre- and post-intervention, except when normality assumptions were violated. In the case of walking velocity, affected leg step time, and hip extension, a Wilcoxon signed rank test was used to compare differences pre- and post-intervention. A two-sided *p*-value of <0.05 was considered statistically significant. When significant differences were found, Cohen’s d was calculated to determine the effect size. A Pearson’s correlation was used to assess how well the number of hours on the AccesSportAmerica Gait Trainer associated with the change in walking speed. 

## 3. Results

From pre- to post-intervention, there was an average 39% increase in walking speed, from 0.65 (SD = 0.44) m/s to 0.84 (SD = 0.50) m/s (Table 1), which was statistically and clinically significant (*p* = 0.016; *d* = 0.40) [21]. When analyzing walking speed together with step length and step time of affected leg and unaffected leg, it was determined that the increase in walking speed from the intervention (training with the AccesSportAmerica Gait Trainer) was due largely to a decrease in step time (affected: *p* = 0.011, *d* = 0.37; unaffected: *p* = 0.004, *d* = 0.67). No significant differences were found in joint angles after the training (Table 2), and no correlation was found between hours spent on the AccesSportAmerica Gait Trainer and the change in walking speed.

## 4. Discussion

Study participants increased their walking speed by an average of 0.19 m/s. An increase in speed of 0.10 m/s is considered a substantial meaningful change in physical performance for community-dwelling older people and subacute stroke survivors [22]. Five participants had pre-intervention walking speeds lower than 0.40 m/s, which would categorize them as home ambulators [23]. Yet, only one of these participants was able to walk for ten meters without personal assistance prior to participating in the intervention. Participant 3 not only increased his assisted walking speed to the category of limited community ambulatory, but he was also able to walk unassisted after the intervention. This participant completed the most sessions (71 sessions) with the AccesSportAmerica Gait Trainer and had the highest average time (M = 43 min) on the AccesSportAmerica Gait Trainer per session. Similarly, Participant 4 improved his walking speed with a cane and was more comfortable walking without it post-intervention. Two participants were categorized as limited community ambulators (from 0.40 to 0.80 m/s) pre-intervention, one of whom improved sufficiently post-intervention to be categorized as a community ambulator (above 0.80 m/s) [23]. The remaining four participants were categorized as community ambulators pre-intervention, with the fastest walker being the only study participant to have a decreased walking speed post-intervention. 

The AccesSportAmerica Gait Trainer elicited an increase in walking speed during overground walking as a result of increased cadence. Eight of the 10 participants who increased their walking speed had at least a 10% decrease in stride time. While not a significant change in this group of participants, stride length was a factor for many participants in the program. Six of the eleven participants increased the step length of both limbs by over 10% post-intervention, with an additional participant increasing the step length of his affected limb only by 20%. The stride length symmetry, defined as the ratio of stride length between the unaffected and affected limbs, improved for eight participants.

Although the diagnoses of the research participants were heterogeneous, common impairments included increased tone, limited range of motion, and decreased motor control. The activity limitation of decreased ambulation was also shared. The degree of their impairments and functional abilities varied, as indicated by walking speed—the functional vital sign. The diversity of the participants supports the conclusion that the AccesSportAmerica Gait Trainer may provide an effective intervention for a broad range of diagnoses and functional limitations.

Changes in walking technique differed between participants. Therefore, impacts of training with the Gait Trainer on joint range-of-motion cannot be generalized between participants. Rather, individual changes in joint angles can be assessed based on participant function. A limitation of this study was that out-of-plane motion influenced the measurement of sagittal joint angles from the video recordings. For this population, out-of-plane motion is much greater than in healthy walkers. Future studies will involve the use of a biomechanics laboratory with 3D motion capture to analyze joint kinematics pre- and post-intervention, which will allow assessment of joint angle through all degrees of freedom. 

The use of a biomechanics laboratory in future studies will also make the measurement of ground reaction forces and the calculation of joint work possible. It is known that the capacity to increase walking speed is limited by impaired hip and ankle power generation in lower-functioning persons [24]. Therefore, the inability to calculate power generation without inground force plates limits our ability to interpret changes in joint angles. For example, one would expect that an increase in hip extension or ankle plantarflexion at toe-off would indicate greater muscle moments at these joints. However, a participant who is actively pushing against the ground, rather than simply lifting his or her leg, may exhibit a decrease in passive extension that is a result of inertial responses to gravity. Therefore, the use of ground reaction forces is necessary to properly interpret changes in joint motion. 

Another future direction for this study is to assess the health benefits of the AccesSportAmerica Gait Training program. Many participants in the program cannot walk without assistance. The lack of physical activity in sedentary wheelchair users results in poor cardiometabolic risk profiles, low self-esteem, social isolation, and depression [25]. The predominant mode of physical activity for wheelchair users, which has been shown to improve cardiometric profiles, is through upper-limb exercises. The AccesSportAmerica Gait Trainer not only allows aerobic exercise, but also provides active resistance training for the lower body. Heart rate monitors can be worn by participants during training sessions to track the magnitude of aerobic exercise and the 20-Item Short Form Survey can be employed pre- and post-intervention to evaluate the program impacts on community participation and quality of life [26]. 

## 5. Conclusions

The AccesSportAmerica Gait Trainer has the potential to improve the walking speed of people with mobility disorders. Due to its body weight-support system and ease of adjustment, the device can be used by patients with a wide range of function. Patients with brain injury who participated in a training program on the AccesSportAmerica Gait Trainer increased their walking speed, largely due to an increase in cadence. Follow-up studies involving a larger population will assess the impact of this intervention on joint work during gait, cardiovascular health, and quality of life.

## Figures and Tables

**Figure 1 medicina-55-00420-f001:**
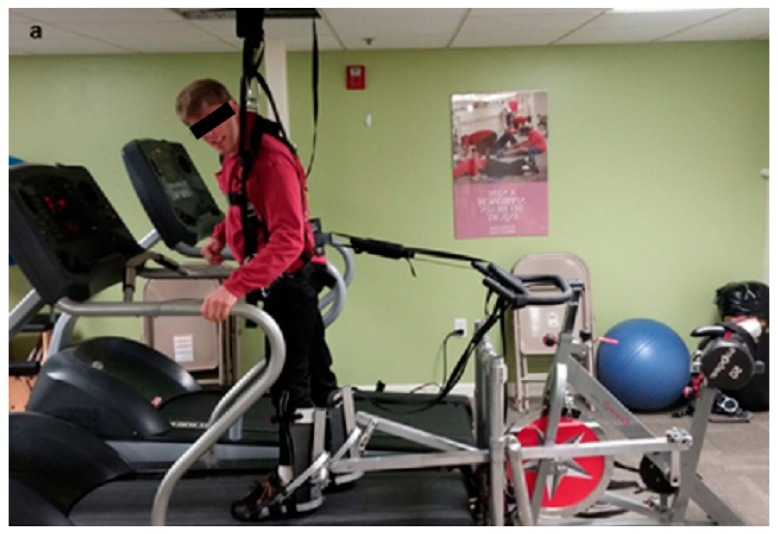
(**a**) A participant in the AccesSportAmerica Gait Trainer Program, supported by an overhead harness system and strapped into the boots of the Gait Trainer. (**b**) Schematic of the AccesSportAmerica Gait Trainer, as installed on a treadmill [16]. The participant steps onto the treadmill and a monitoring trainer helps strap the participant into the harness and boots. The motion is provided by the moving treadmill, aided by the participant (or a trainer pedaling the bicycle if necessary), with the momentum of the flywheel and the adjustable shafts providing smooth and perfect form to the step cycle.

**Table 1 medicina-55-00420-t001:** Training information (number of weeks in the program, total number of sessions, and average time on the AccesSportAmerica Gait Trainer per session) and pre- and post-intervention walking velocity (m/s) and step length (m) and time (s) of the affected and unaffected limbs. The participants are presented in order of pre-intervention walking velocity. 10MWT: Ten Meter Walk Test; CP: cerebral palsy; TBI: traumatic brain injury.

Partic.	Diagnosis	Training			10MWT		Affected Leg				Unaffected Leg			
		Weeks	Sessions	Time, min	Velocity, m/s		Step Length, m		Step Time, s		Step Length, m		Step Time, s	
					Pre	Post	Pre	Post	Pre	Post	Pre	Post	Pre	Post
1	Stroke	40	50	31	0.23	0.31	0.54	0.64	0.63	0.62	0.64	0.58	1.03	0.97
2	TBI	30	23	39	0.25	0.36	0.37	0.42	2.00	1.50	0.43	0.48	0.97	0.87
3	TBI	34	71	43	0.29	0.54	0.42	0.38	3.00	1.83	0.44	0.43	1.43	1.07
4	Stroke	29	27	31	0.30	0.39	0.38	0.48	1.37	1.00	0.30	0.39	1.03	0.70
5	CP	35	70	42	0.33	0.56	0.56	0.47	1.07	0.77	0.50	0.56	1.40	1.07
6	Stroke	28	20	33	0.50	0.88	0.42	0.58	0.73	0.67	0.34	0.72	0.93	0.83
7	Stroke	17	17	35	0.58	0.65	0.41	0.53	0.87	0.77	0.44	0.53	0.70	0.60
8	TBI	16	18	28	0.95	1.34	0.48	0.47	0.47	0.33	0.41	0.42	0.83	0.33
9	Stroke	12	12	25	1.13	1.15	0.73	0.69	0.63	0.63	0.55	0.52	0.53	0.53
10	Stroke	35	56	41	1.29	1.89	0.67	0.85	0.40	0.37	0.51	0.71	0.60	0.43
11	Stroke	35	42	32	1.33	1.15	0.68	0.82	0.47	0.50	0.73	0.81	0.77	0.77
Mean		29.8	36.9	34.5	0.65	0.84	0.51	0.58	1.06	0.82	0.48	0.56	0.93	0.74
SD		8.9	21.8	6.0	0.44	0.50	0.13	0.16	0.80	0.47	0.13	0.14	0.29	0.25
				*p*-value		0.016	0.052 -		0.011 0.37		0.059 -		0.004 0.67	
				Cohen’s d		0.40								

**Table 2 medicina-55-00420-t002:** Affected side and walking aid of each participant and joint angles (degrees) of the affected leg pre- and post-intervention. Ankle dorsiflexion (DF) was measured at toe-off of the contralateral leg; ankle plantarflexion (PF), knee flexion, and hip extension were measured at toe-off; and hip flexion was measured at foot strike. AFO: ankle foot orthosis; biAFO bilateral ankle foot orthosis.

Partic.	Affected	Walking	Ankle DF		Ankle PF		Knee Flexion		Hip Extension		Hip Flexion	
	Side	Aide	Pre	Post	Pre	Post	Pre	Post	Pre	Post	Pre	Post
1	right	assisted	6	7	8	6	42	46	25	31	18	36
2	left	assisted	9	−7	−34	−26	44	33	25	27	40	39
3	left	assisted/AFO	−9	−4	−36	14	30	13	−2	7	19	8
4	right	cane/AFO	−16	−15	−39	−30	19	14	2	2	4	16
5	left	assisted/biAFO	2	7	−18	−8	44	47	−3	−6	8	15
6	right	none	5	12	−10	−18	48	38	5	−4	30	30
7	right	none	5	−2	−9	−9	42	39	4	−3	17	23
8	right	biAFO	11	−4	6	−9	33	28	7	−8	19	22
9	right	AFO	−4	5	−26	−21	13	21	−13	2	27	22
10	left	AFO	22	1	−26	−10	29	18	−19	−13	−6	17
11	right	AFO	−3	−11	−23	−28	35	30	−16	−17	20	16
		Mean	0.3	−1.5	−18.1	−12.1	34.3	29.9	1.5	5.2	18.7	21.1
		SD	9.1	8.4	15.9	13.1	11.4	12.8	14.5	13.9	10.7	10.3
		*p*-value		0.297		0.268		0.059		0.477		0.195

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
