# Peer review of "The Effect of a Low-Cost Body Weight-Supported Treadmill Trainer on Walking Speed and Joint Motion"

_medicina, 2019, doi:10.3390/medicina55080420_

Round 1
Reviewer 1 Report
Dear authors,
thank you to give me the opportunity to review this study.
The main topic of the article is relevant. Devices for deambulation represent an important issue in rehabilitation.
Unfortunately, this study is a single-arm study, without control group. Sample size is limited, with patients affected by different diseases. Some medical and rehabilitative issues are not discussed (i.e. spasticity. Have patients similar degree of spasticity? Have you assessed it through dedicated evaluation scales?).
Moreover, you reported "affected" and "unaffected" lower limb. In stroke patients this concept is simple. Are TBI and CP patients clinically hemiplegic?
About methods, I have some doubts about statistical methods adopted to analyze data. You have mean values with high SD. This is normal in small sample size. Just as example, in 10MWt pre you have mean 0.65 and SD 0.44. These data indicate that data are not normally distributed!. In this case and with 11 subjects, it is difficult to consider parametric test as paired t-test. Probably, a non-parametric approach is more adapt for your data. Re-evaluate Pearson correlation too.
Probably, some significant results could be different with this approach, but using parametric tests violates many statistical assumptions.
Please, use common ID for patients in table 1 and 2. You stated that five patients had pre-intervention walking speeds lower than 0.40 m/s (patietns 1 to 5 in table 1) but only one was able to walk without assistence (this statement is not in line with the first five subjects in table 2).
Labels in lower panel of figure 1 are not useful. Please, try to specify better how the patient uses the AccesSportAmerica Gait Trainer device.
Author Response
Thank you for the confirmation as to the relevance and importance of this study. We greatly appreciate your constructive comments. The changes we have made to the manuscript in response have made it stronger and clearer, for which we are very grateful.
We agree that the small sample size and lack of a control group are limitations of the study. Recruitment was limited by the participants enrolled in the Gait Trainer Program. We are seeking funding to be able to recruit a larger pool of participants. The use of a control group in future studies is an excellent suggestion.
We did not evaluate levels of spasticity or tone of our participants. However, we have added the following description of the participants to the methods section: “All conditions presented with high tone and affected the motor control of the extremities and trunk.” (84-85) We also added an explanation as to how the affected limb was defined: “When pathologies were bilateral, the weaker limb was defined to be the affected limb for purpose of analysis.” (87-88)
At your suggestion, we tested for normality using the Shapiro-Wilk test and substituted the Wilcoxon Signed Rank Test when normality assumptions were violated. Regarding the Pearson correlation, time on the Gait Trainer and changes in walking velocity were normally distributed; therefore this statistical test was appropriate. We updated the affected p-values in the results and edited the methods section to describe the change in statistical analyses:
Statistical analyses were performed using the software package SPSS 25.0. Normality of the data was tested using the Shapiro-Wilk test. A paired Student’s t-test was used to compare differences pre- and post-intervention, except when normality assumptions were violated. In the case of walking velocity, affected leg step time, and hip extension, a Wilcoxon Signed Rank test was used to compare differences pre- and post-intervention. (134-139)
The Subject IDs match in Tables 1 and 2. We added the header “Participant #” to both tables to clarify the tie between the two data sets and the sentence “The participants are presented in order of pre-intervention walking velocity”to the Table 1 caption. We double-checked the consistency between the written text and the presentation of data in the tables and are satisfied that they match. To clarify that “assistance” does not include devices, the adjective “personal” was added: “Yet, only one of these participants was able to walk for ten metres without personal assistance prior to participating in the intervention.” (165-167)
At your suggestion, we removed the labels from the schematic of the Gait Trainer in Figure 1. We also expanded the figure caption to better define what is being shown:
Figure 1. (a) A participant in the AccesSportAmerica Gait Trainer Program, supported by an overhead harness system and strapped into the boots of the Gait Trainer. (b) Schematic of the AccesSportAmerica Gait Trainer, as installed on a treadmill [16]. The participant steps onto the treadmill and a monitoring trainer helps strap the participant into the harness and boots. The motion is provided by the moving treadmill, aided by the participant (or a trainer pedaling the bicycle if necessary), with the momentum of the flywheel and the adjustable shafts providing smooth and perfect form to the step cycle. (93-99)
In addition, we expanded the description of the Gait Trainer in the text:
Installed posterior to a standard treadmill with an overhead harness system, the AccesSportAmerica Gait Trainer guides users through the walking cycle by effectively linking the external motion of the treadmill belt into the body’s own kinetic chain (Figure 1)[16]. Support and precise motion are delivered to the lower leg and foot through a boot that is attached to a flywheel via fully-adjustable shafts. The micro adjustments available on the AccesSportAmerica Gait Trainer allows for precise fitting to the participant’s size and easy variation of stride length, hip and knee flexion, and ankle rotation and heel-strike angle. The angular momentum contributed by the coupled flywheel facilitates smooth continuous motion through the entire gait cycle. The device requires only one monitoring trainer who can also pedal the stationary bicycle to influence gait, if necessary. The AccesSportAmerica Gait Trainer speeds can vary from 1.2 to 2 mph or 2 to 6 mph, depending on the stride length. The training device is versatile as it allows users to walk and even run safely. Participants can train for over an hour at a time for benefits of aerobic conditioning and improving biomechanical function. The AccesSportAmerica Gait Trainer avoids costly electronics, hydraulics, and other complex systems by employing a simple, robust design. Due to its simplicity, the device may be produced at lower cost, making it affordable to more facilities and accessible to a greater number of persons living with ambulatory difficulty. (55-70)
Reviewer 2 Report
The authors report on a new type of walker which allows to measure walking speed and joint angles and to assess changes of these parameters after training. Walkers are usually expensive, which prevents a wide spread use of the walkers. The AccesSport America Gait Trainer is cheaper but has some disadvantages. Out of plane movements and ground reaction forces cannot be measured. In future, there will be additional devices which will be combined with the AcessSportAmerica Gait Trainer.
The authors should explain in some more details the technical differences between the expensive walkers and the AccesSport America Gait Trainer and why the new walker is much cheaper.
The group of participants was very small and heterogenous. Since this was a study which was more like a proof of principle, one can consider the heterogenousity of the group as an advantage. Futher studies may work out for whom the new walker is most suitable. The topic is interesting and very important for rehabilitation, Therefore, the study should be published with minor changes (comparison between the walkers)
Author Response
Thank you for the confirmation as to the relevance and importance of this study. We greatly appreciate your constructive comments. The changes we have made to the manuscript in response have made it stronger and clearer, for which we are very grateful.
First, we would like to clarify that the limitations regarding out-of-plane movement and ground reaction forces that are presented in the Discussion section are not made in regards to the Gait Trainer. Rather, these limitations are indicative of the methods used in the current study, particularly of employing a video camera rather than 3D motion capture in a Biomechanics Laboratory equipped with installed force plates. We are pursuing the means to conduct such a study, with more sophisticated data analysis techniques.
To address your comment regarding comparisons between the Gait Trainer and other options, we expanded the description of the Gait Trainer in the text:
Installed posterior to a standard treadmill with an overhead harness system, the AccesSportAmerica Gait Trainer guides users through the walking cycle by effectively linking the external motion of the treadmill belt into the body’s own kinetic chain (Figure 1)[16]. Support and precise motion are delivered to the lower leg and foot through a boot that is attached to a flywheel via fully-adjustable shafts. The micro adjustments available on the AccesSportAmerica Gait Trainer allows for precise fitting to the participant’s size and easy variation of stride length, hip and knee flexion, and ankle rotation and heel-strike angle. The angular momentum contributed by the coupled flywheel facilitates smooth continuous motion through the entire gait cycle. The device requires only one monitoring trainer who can also pedal the stationary bicycle to influence gait, if necessary. The AccesSportAmerica Gait Trainer speeds can vary from 1.2 to 2 mph or 2 to 6 mph, depending on the stride length. The training device is versatile as it allows users to walk and even run safely. Participants can train for over an hour at a time for benefits of aerobic conditioning and improving biomechanical function. The AccesSportAmerica Gait Trainer avoids costly electronics, hydraulics, and other complex systems by employing a simple, robust design. Due to its simplicity, the device may be produced at lower cost, making it affordable to more facilities and accessible to a greater number of persons living with ambulatory difficulty. (55-70)
Finally, we appreciate your comment regarding the heterogeneity of the participants in this study. We thought it would be a good idea to include such commentary in the discussion and have added the following paragraph:
Although the diagnoses of the research participants were heterogeneous, common impairments included increased tone, limited range of motion and decreased motor control. The activity limitation of decreased ambulation was also shared. The degree of their impairments and functional abilities varied, as indicated by walking speed, the functional vital sign. The diversity of the participants supports the conclusion that the AccesSportAmerica Gait Trainer may provide an effective intervention for a broad range of diagnoses and functional limitations. (184-189)
Round 2
Reviewer 1 Report
Dear Authors,
thank you for changes applied to the manuscript.
The figures and tables have been improved consistently, as the statistical section too.
I appreciate your efforts.
Thank you